# Neonatal Neuroimaging in Neonatal Intensive Care Graduates Who Subsequently Develop Cerebral Palsy

**DOI:** 10.3390/jcm11071866

**Published:** 2022-03-28

**Authors:** Malcolm R. Battin, Sîan A. Williams, Anna Mackey, Woroud Alzaher, Alexandra Sorhage, N. Susan Stott

**Affiliations:** 1Newborn Services, Auckland City Hospital, Park Road, Grafton, Auckland 1023, New Zealand; 2Liggins Institute, University of Auckland, Park Road, Grafton, Auckland 1142, New Zealand; sian.williams@auckland.ac.nz; 3New Zealand Cerebral Palsy Register, Starship Child Health, Auckland 1023, New Zealand; amackey@adhb.govt.nz (A.M.); worouda@adhb.govt.nz (W.A.); asorhage@adhb.govt.nz (A.S.); 4Department of Paediatric Orthopaedics, Starship Child Health, Auckland 1023, New Zealand; s.stott@auckland.ac.nz; 5Department of Surgery, University of Auckland, Auckland 1023, New Zealand

**Keywords:** cerebral ultrasound, magnetic resonance imaging, cerebral palsy

## Abstract

Cerebral palsy is a common cause of physical disability. The New Zealand Cerebral Palsy Register (NZCPR) was established in 2015 and reports national data. Internationally, an early CP diagnosis has been a focus, with imaging and clinical tools used to enable early accurate detection. Accordingly, guidelines are being developed for New Zealand, including a specific pathway for high-risk neonatal intensive care (NICU) graduates, reflecting the high rate of CP in this group. To inform this work, we reviewed imaging data from a retrospective NICU cohort identified from the NZCPR. In these 140 individuals with CP and a confirmed NICU admission during 2000–2019 inclusive, imaging frequency, modality, and rate of abnormality was determined. Overall, 114 (81.4%) had imaging performed in the NICU, but the frequency and modality used varied by gestational subgroup. For infants born at less than 32 weeks gestation, 53/55 had routine imaging with ultrasound, and IVH was graded as none or mild (grade 1–2) in 35 or severe (grade 3–4) in 18 infants. For the 34 infants born between 32–36 weeks gestation, only 13/19 imaged in the NICU were reported as abnormal. For 51 term-born infants, 41/42 imaged in the NICU with MRI had abnormal results.

## 1. Introduction

Cerebral palsy (CP) is a common cause of physical disability, which majorly impacts affected children and their families [1]. The prevalence is typically reported as approximately 2.1 per 1000 live births in high-income countries [2], but recent Australian data documents a decrease in rate [3,4]. Locally, the New Zealand Cerebral Palsy Register (NZCPR) was established in 2015. It facilitates benchmarking with overseas experience and promotes opportunities for research. One focus of recent benchmarking has been the (mean) age at CP diagnosis and concomitant use of assessment tools. The age of CP diagnosis is quite variable across different countries, health systems, and epochs, ranging from 8 to 24 months in recent literature [5]. In New Zealand, over half of families (59%) receive the diagnosis after 12 months of age, and only 13% are diagnosed within the first 6 months (personal communication NZCPR unpublished). In contrast, approximately a quarter of infants are diagnosed before 6 months in Denmark (22%) [6] and Australia (26%) [7]. An earlier diagnosis is appreciated by families [8,9] and, due to early life neuroplasticity, provides an opportunity for optimal response to intervention [10]. Building on the evidence that imaging and clinical assessment tools enable early detection of CP with a high level of accuracy (90–95%) [11,12,13], the implementation of guidelines facilitating earlier CP diagnosis has proven successful overseas [14] and is underway in New Zealand [15]. 

An association between neonatal unit admission and CP is recognised [16] and warrants development of a specific pathway for assessment of high-risk neonatal intensive care (NICU) graduates. Conditions such as extreme prematurity [17] and neonatal encephalopathy (NE) are particularly associated with an increased risk of CP [18,19]. Moreover, a novel follow-up clinic for NICU graduates reported an encouraging preliminary experience [20] using proven imaging and clinical assessment tools [11,12,13] to optimise early CP detection. Thus, review of the existing local NICU imaging practice with respect to CP is timely to inform ongoing work.

The use of cerebral ultrasound (CUS) and Magnetic Resonance Imaging (MRI) to predict CP has been subject to systematic review [12]. However, there is a paucity of data on the use of imaging during NICU stay in a New Zealand context, particularly as use varies depending on gestational age (GA) and clinical condition. As MRI (or previously CT) will be typically used in term infants and a standardised program of bedside CUS is used in preterm infants, it is important to review yield by gestational age group. Finally, study of available ethnicity data will support an equitable access. Accordingly, this study reviewed data from a retrospective ADHB NICU cohort identified from the NZCPR. The specific study objectives were:

(1)To determine the frequency and modality of neonatal cerebral imaging used in NZ for high-risk infants who subsequently developed CP, including sub-analysis by GA and ethnicity, plus benchmarking against available guidelines.(2)To document the rate of abnormality and characterise CP diagnosis (including functional and topographical classifications) in relation to abnormal neuroimaging in this NZ-specific cohort.

## 2. Materials and Methods

### 2.1. Participants

The Auckland District Health Board (ADHB) NICU is one of six Level 3 NICUs in New Zealand with an average of 800–1000 admissions per year, of whom approximately 150–200 weigh less than 1500 g [21]. Co-located with Starship Children’s Hospital, the ADHB NICU is also the referral centre for paediatric cardiology and neonatal surgery.

In the NZCPR dataset, 44% of children have been identified to have a history of admission to a NICU or SCBU. Thus, to obtain a representative cohort of children with a diagnosis of CP-associated neonatal admission, we identified ADHB NICU graduates from the NZCPR born 2000–2019 inclusive. ADHB NICU admissions include a cross section of gestational ages, with approximately 200 babies per year born before 32 weeks, 250 born at 32 to 36 weeks, and more than 400 term infants. A 20-year epoch was selected to have an adequately sized cohort of NICU graduates with reasonably consistent imaging practice, including a standardized CUS program and available 1.5 T/3 T MRI (Siemens Healthcare, Erlangen, Germany).

The study was conducted in accordance with the Declaration of Helsinki, and Ethical Approval for use of the NZCPR data was under Health and Disability Ethics Committee approval 13/NTA/130/AM16, to collect health information, including imaging, for children and adults with CP.

### 2.2. Data Capture

The initial NZCPR dataset, including date of birth, sex, CP severity distribution/type, and age of diagnosis, was supplemented with clinical information from available DHB sources. Gestation and birth weight were confirmed from DHB data sources, and then additional data were extracted from the neonatal database and electronic reports regarding services received including neonatal imaging. Ethnicity was defined using that identified from the medical record, including Māori, Pacific; Indian; Chinese/other Asian; MELAA; NZ European; and Other/Other European. The unit practice guideline [22] was for CUS to be performed around day 5, day 28, and 36 weeks corrected gestation for infants born below 30 weeks gestation or below 1250 g birth weight.

### 2.3. Data and Statistical Analysis 

A descriptive analysis was performed using three predefined gestational age bands and by ethnicity. Division by gestational age was: (1) extremely or very preterm infants born at less than 32 weeks gestation; (2) moderate to late preterm infants born between 32–36 weeks gestation; and (3) infants at term (37–42 weeks gestation). Benchmarking ADHB NICU practice to available guidelines [22] was undertaken. Specifically, infants born below 32 weeks’ gestation received Level 3 neonatal intensive care based on gestation and qualified for a standardised screening protocol using CUS [22]. In contrast, imaging in moderate to late preterm infants was based on the attending team practice and clinical situation. For term infants, consensus guidelines exist for MRI following NE [23], but other conditions were investigated on an individual basis.

Data are presented as mean (SD) unless not normally distributed, when median (IQR/range) is used. For analysis by pre-specified subgroups and ethnicity, categorical data were analysed using Chi square and Fischer’s exact test as appropriate. 

## 3. Results

A study cohort of 140 individuals with CP diagnosis and ADHB NICU admission was identified from the NZCPR for the period 2000–2019 inclusive. For this cohort, the median (range) gestation age at birth was 33 (23–42) weeks and median birth weight was 1998 (700–5090) g. Ethnicity was recorded as Māori in 22 (15.7%), Pacific in 19 (13.1%), Indian in 7 (4.6%), Chinese/other Asian 4 (2.6%), MELA 2 (1.3%), NZ European 75 (55.6%), and Other European 11 (7.2%). The median length of stay in NICU was 30 (range 1–189) days.

The type/distribution of CP assigned at five years was documented in the NZCPR for 137 individuals. Overall, the most common distribution pattern was spastic diplegia in 51 (36.4%), then spastic quadriplegia in 39 (27.8%), followed by left hemiplegia/monoplegia in 22 (15.7%), and right hemiplegia/monoplegia in 16 (11.4%). There were also a smaller number with less common distributions, including three (2.1%) with Spastic Triplegia; three (2.1%) Dyskinesia, mainly Dystonia; one each (0.7%) hyptonia, Unknown/unclassified and high risk for CP not confirmed yet due to age. The severity of CP, assigned at 5 years, was graded by Gross Motor Function Classification System (GMFCS) with a greater number at the milder levels: 43 (30.7%) Level I, 38 (27.1%) Level II, 13 (9.3%) Level III, 27 (19.3%) Level IV, 14 (10.0%) Level V, and 5 (3.6%) unknown.

The cohort included: 55 (39.3%) infants born under 32 weeks’ gestation; 34 (24.3%) moderate to late preterm infants born 32–36 weeks gestation; and 51 infants (36.4%) born at term. The demographic and clinical details are summarized in Table 1.

Overall, the number of individuals with imaging specifically performed during the neonatal period was 114/140 (81.4%), but imaging performed, including modality, varied for each gestational subgroup. An additional 16 infants were imaged after NICU discharge, so 130 (92.8%) of the cohort underwent cerebral imaging prior to diagnosis of CP. The details of imaging are summarized in Table 2.

### 3.1. Extremely or Very Preterm 

Of the 55 extremely or very preterm infants born before 32 weeks’ gestation, 53 had routine imaging with CUS. The two infants who did not have ultrasound performed were both born at 30 weeks gestation and weighed more than 1250 g. Early ultrasound intraventricular haemorrhage (IVH) was graded as ‘none’ in 23 or ‘mild’ (Papile 1–2) in 12 and ‘severe’ (Papile grade 3–4) in 18 infants. On the assessment by late CUS, there was documented white matter injury in 5/35 (normal or mild IVH) and 5/18 (severe IVH), respectively. Thus, 23 infants had either severe IVH or documented white matter injury. One (29 week GA, 1185 g) infant underwent imaging with MRI to confirm cerebral dysplasia, in addition to the routine CUS. For this subgroup, there was 100% accordance with the recommended practice of cerebral ultrasound screening for infants below 30 weeks gestation or below 1250 g birth weight. However, a high proportion 30/53 (57%) had either no IVH or mild (Papile grade 1–2) IVH.

### 3.2. Moderate to Late Preterm 

For the 34 infants born at 32–36 weeks gestation and diagnosed with CP, 19 were imaged in the NICU following birth and 18 were initially imaged with CUS, with five having subsequent neonatal MRI and one CT (early in cohort) performed to further document injury. One further infant born weighing >1500 g had an MRI but no CUS in NICU. Thirteen babies were reported to have abnormal imaging, including eight with significant white matter injury, two with major haemorrhage plus one each congenital anomaly due to toxoplasma infection, antenatal, cyst and basal ganglia damage. In each case, imaging appeared to be based on individual clinical situation, with no applicable guideline.

### 3.3. Term

For 51 term-born infants subsequently diagnosed with CP, 42 were imaged in the NICU with MRI and 41 were reported as abnormal results, including a selection of aetiologies and clinical presentations. Congenital abnormalities were found in 10 (24.4%) scans, including: Dandy Walker malformation (*n* = 2); Hemimegalencephaly (*n* = 2); congenital hydrocephalus (*n* = 2); microcephaly (*n* = 1); and chromosomal abnormality associated with frontoparietal and perisylvian polymicrogyria (*n* = 1) plus antenatal cystic/hydrocephalus changes (*n* = 2). These are classified as A (i.e., maldevelopments) in the harmonised MRI classification scheme [24]. Two other common clinical diagnoses were hypoxic ischaemic encephalopathy in 20 (48.8%) with extensive damage of basal ganglia/deep grey matter in 17/20, and middle cerebral artery infarction in six (14.6%); both are classified under C (predominant grey matter injury) in the harmonised MRI classification scheme. The remaining five individuals displayed variable degrees of white matter injury, classified under B (predominant white matter injury), and subdural haemorrhage from a variety of aetiologies. Judging against the standard for NE, which had a relevant guideline, there was 100% compliance with MRI in a timely manner.

The distribution of CP by gestational age is shown in Figure 1. Diplegia was more common in preterm groups (*p* = 0.002). In addition, the small number of term infants with diplegia displayed WM damage on imaging. On review of CP severity, it was documented that for the 30 infants born <32 weeks without evidence of severe IVH or white matter injury on CUS, 26 (86.7%) were reported to be classified GMFCS level I or II, reflecting a milder degree of CP. In addition, the three preterm babies with GMFCS level V at 5 years all had extensive severe IVH.

Cerebral imaging provision for the three largest ethnic groups was 64/75 (73%), 20/22 (91%), and 14/19 (74%) for NZ European, Māori, and Pacific, respectively. Similarly, abnormality was detected in 53/64 (83%), 14/20 (70%), and 12/14 (86%) for NZ European, Māori, and Pacific, respectively. There was no statistically significant difference in proportion with imaging performed (*p* = 0.297) or rate of abnormal imaging (*p* = 0.392) analysed by ethnicity.

## 4. Discussion

Review of this cohort confirmed that neonatal neuroimaging was performed in over 80% and documented the typical conditions seen in NICU graduates. Although these relatively high imaging rates were encouraging, opportunities exist to increase the utility of the information obtained apropos developing CP. In addition, awareness of the variation in both imaging frequency and modality across the three gestational age groups is important for planning strategies to optimise CP diagnosis.

Neonatal ultrasound for very preterm infants is a standardised program that includes both early scans around five days, when most cases of IVH are visible [25], and later imaging to detect white matter injury. Although there was a high compliance with the program, normal or only mildly abnormal ultrasound results were common in this group, so would not have triggered focused follow up for CP. This is consistent with the lower sensitivity of ultrasound compared with MRI [12] and means that supplemental MRI may be indicated in some infants. As neonatal intensive care in NZ is provided in six Level 3 units with equivalent ultrasound programs, this experience is reasonably generalisable and suggests consideration should be given to selective supplemental MRI.

Imaging use in the moderate to late preterm 32–36 week group was variable, with low utilisation but a moderate (68.4%) rate of abnormality. Although this lower utilisation may present an opportunity for increase, the group typically have lower rates of CP, plus more babies are born in this group than extremely preterm infants. As this gestation group accounts for about 17 percent of all children with CP [4], delineating any potential benefit is important. Neuroimaging in late and moderate late preterm infants has recently been reviewed [26,27], with varying rates of abnormality reported. Currently, there is insufficient evidence to recommend CUS as a routine screening tool for moderate late prematurity [28]. Retrospective cohort studies suggest that the yield may be increased by the use of risk categories based on neurological examination, Apgar score, head circumference, or requirement for ventilation or vasopressor drugs [28,29,30]. However, there is a need for further prospective study and correlation with neurodevelopmental outcomes [28].

Two recent publications regarding unselected MRI use further inform practice. The first reports MRI in an unselected cohort of infants born below 33 weeks gestation to be 17% sensitive and 94% specific for a normal motor outcome [31]. The second, a recent APA statement [32] states that MRI < 30 weeks is not indicated as routine, but may be offered at term equivalent age to high-risk infants. Unfortunately, there is currently no clear consensus on which babies to select for MRI at term. This would benefit from further study, particularly with reference to generating local data from the NZ system.

For the term infants there was a variety of clinical conditions but over 80% imaging uptake. NE investigation, guided by a consensus document recommending imaging [23], had excellent uptake. However, other conditions without imaging protocols also had high uptake. For this group, the imaging was with MRI and the rate of abnormal imaging was high, so the experience appears similar to that in other ages with respect to utility in diagnosis of CP [11,12]. MRI is available in other tertiary neonatal centres in New Zealand, so our findings are likely generalisable. However, it is not an unlimited resource and agreed guidelines would assist in directing use.

The good adherence to guidelines for preterm infants and term infants with NE is encouraging but not surprising given the clinical experience and underpinning evidence base. The strong relationship between diplegia and prematurity contributes further to this evidence base. However, it was also confirmed that preterm infants with CP may have normal CUS imaging and require complementary clinical approaches [11,12,13] to detect CP early. As further guidelines are developed pertaining to moderate and late preterm infants, ongoing research should be incorporated, including health services perspectives and economic evaluation. The local data on CP by gestational age for NICU graduates can inform screening programs and be used in modelling alternative methods for early CP diagnosis such as General Movements (GM) and Hammersmith Infant Neurological Examination (HINE) [12,13]. This is important in planning NZ services, particularly if access to MRI is limited by cost or availability. Finally, a small number were considered to have a probable postnatal origin to CP, with changes in their imaging findings reflecting events outside the neonatal period including trauma and major surgery.

It is important to review service provision for any potential disparities between Māori and non-Māori. Although there was not a difference seen in the provision and outcome of cerebral imaging by ethnicity, it is vital that any new approaches to care do not impact adversely or increase disparity. This is recognised in planning the pathway for diagnosis in high-risk infants.

The study strengths include the use of data from NZCPR, which has standardised data fields. Together with the DHB records these formed an excellent data source, particularly as both datasets have undergone prior scrutiny and cleaning. However, a study weakness was that the NZCPR requires case registration, and imaging was not reported in a standardised manner. Although reports typically documented the distribution and severity of any observed changes, often with comment on the aetiology, the format was variable. Likewise, the 20-year epoch encompassed significant practice change including improvements in imaging quality, plus imaging technique may have an element of operator dependency and pick up may vary between modalities. Finally, the dataset only includes individuals with a documented diagnosis of CP so cannot be used to study the ability to make a diagnosis of CP, which comes from other publications [11,12]. The planned national introduction of high-risk pathway in NZ will provide opportunities to further assess and refine the system, including the acquisition of data on imaging, rates of completion of screening tests, and detection rate from other assessments such as GM and HINE.

## 5. Conclusions

Overall, this research has identified four key areas for development. First, increased uptake of imaging for those born at 32–36 weeks. Ideally this would be informed by a guideline including useful selection criteria. Second, development of criteria for supplemental MRI, at term equivalent GA, following completion of CUS in preterm infants. Third, standardised reporting as per the published framework [24], which enables clear communication with family and other services. Finally, a formal plan for the discussion of findings with families, whilst in the NICU, will aid their understanding of long-term consequences.

## Figures and Tables

**Figure 1 jcm-11-01866-f001:**
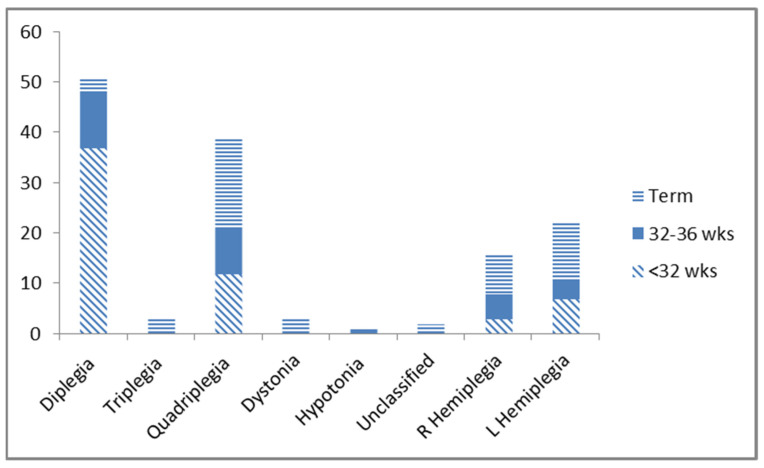
Cerebral Palsy topographic distribution (*Y* axis = number).

**Table 1 jcm-11-01866-t001:** Demographic and clinical details. BW = birth weight, GA = Gestational age, LOS = Length of stay.

	Extremely or Very Preterm <32 Weeks, *n* = 55	Moderate to Late Preterm 32–36 Weeks, *n* = 34	Term 37–42 Weeks, *n* = 51
Mean (SD) GA (weeks)	27.9 (2.4)	33.6 (1.35)	39.2 (1.39)
Mean (SD) BW (g)	1242 (385)	2024.1 (464.8)	3360.2 (585.9)
Male:Female	33 (60%):22 (40%)	18 (53%):16 (47%)	27 (53%):24 (47%)
Mean (SD) LOS (days)	68.2 (40)	31 (21.5)	18.7 (12.8)
GMFCS I–II	37 (67%)	22 (65%)	23 (45%)
GMFCS III–V	18 (33%)	11 (32%) (1 unknown 3%)	24 (47%) (4 unknown 8%)

**Table 2 jcm-11-01866-t002:** Details of imaging.

	Extremely or Very Preterm <32 Weeks, *n* = 55	Moderate to Late Preterm 32–36 Weeks, *n* = 34	Term 37–42 Weeks, *n* = 51
Neuroimaging during neonatal admission	53/55 (96%) *	19/34 (55.9%) *	42/51 (82.3%) *
Abnormal neonatal imaging	30/53 (56.6%)	13/19 (68.4%)	41/42 (97.6%)
Initial imaging performed after NICU discharge	1 (1.8%)	6 (17.6%)	9 (17.6%)
No cerebral imaging prior to CP diagnosis	1 (1.8%)	9 (26.5%)	0 (0%)
Postnatal cause CP	3 (5.45%)	1 (2.9%)	3 (5.88%)

* includes infants with probable postnatal origin CP (e.g., perioperative).

## Data Availability

The data are not publicly available but are held by NZCPR and requests for research access dealt with on an individual basis.

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
