# Peer review of "Neonatal Neuroimaging in Neonatal Intensive Care Graduates Who Subsequently Develop Cerebral Palsy"

_jcm, 2022, doi:10.3390/jcm11071866_

Round 1

Reviewer 1 Report

This study investigated cases of cerebral palsy from the national registration database NZCPR which would have a high capture rate of CP. Therefore, the information is highly accurate and helpful.

Major issue

#1  The survey period are 20 years, and the image resolution of ultrasound and MRI equipment has improved significantly during this 20 years period. Therefore, there may be a change in the detection rate of abnormal findings. I recommend the authors to compare the abnormal imaging rate in the first decade and the last decade of the survey period.

#2   I cannot recognize the details of the abnormal imaging listed in the table 2. Please revise it so that it matches the description in the text. In the text, there are 12 cases of mild IVH and 18 cases of severe IVH, but the number of cases with abnormal imaging is 28 in the table 2.

#3  The large and extensive brain lesions would be associated with severe CP. Therefore, it is more likely that severe CP cases can be detected by imaging study. I recommend the authors to investigate the rate of abnormal imaging by GMFCS level in very preterm subjects.

Minor issue

#4   The MRI equipment tested and its magnetic field strength should be described.

#5   There is a discrepancy in the numbers. Page 3, line 130: 17 infants were imaged after NICU discharge, but in the table 2 they are 16 infants (1 + 6 + 9).

Author Response

Review 1

This study investigated cases of cerebral palsy from the national registration database NZCPR which would have a high capture rate of CP. Therefore, the information is highly accurate and helpful.

We thank the reviewer for this comment. 

Major issue

#1  The survey period are 20 years, and the image resolution of ultrasound and MRI equipment has improved significantly during this 20 years period. Therefore, there may be a change in the detection rate of abnormal findings. I recommend the authors to compare the abnormal imaging rate in the first decade and the last decade of the survey period.

We certainly agree that there have been some improvements in resolution and imaging quality over the last two decades. However, in addressing the comment it might be useful to consider ultrasound and MRI separately. Indeed in the MRI group there was an overall very high 97.6% abnormal imaging rate so not unexpectedly no major difference was present when splitting abnormality rate by decade.

For ultrasound the overall rate for abnormal imaging was 56.6% but it should be noted that the denominator is those infants diagnosed with CP not all NICU admissions. So, detection of smaller or more subtle lesions will only affect this rate if the changes are associated with subsequent CP. Furthermore, there has been a trend to less IVH and less severe IVH over time (Yeo KT et al. Improving incidence trends of severe intraventricular haemorrhages in preterm infants <32 weeks gestation: a cohort study. Arch Dis Child F &N 2020;105:145-150.) On our comparison between the two decades covered by the cohort, we found no statistically significant difference in the number of severe IVH or total abnormal ultrasound scans.

Based on this information no changes were made to the manuscript.  

#2   I cannot recognize the details of the abnormal imaging listed in the table 2. Please revise it so that it matches the description in the text. In the text, there are 12 cases of mild IVH and 18 cases of severe IVH, but the number of cases with abnormal imaging is 28 in the table 2.

Thank you. The table and the text have now been amended. In the table we have now given the total of abnormal images for this group, which is 30 (12+18). Note this number includes grade 1-2 IVH, which would be considered mild. In the text we have also added a sentence to clarify that 23 infants had either a severe grade IVH or documented white matter injury on late scans.   

#3  The large and extensive brain lesions would be associated with severe CP. Therefore, it is more likely that severe CP cases can be detected by imaging study. I recommend the authors to investigate the rate of abnormal imaging by GMFCS level in very preterm subjects.

We have already documented that for the milder degree of imaging abnormalities the majority of the babies have a less severe GMFCS level i.e. the following text on page 5: “for the 30 born <32 weeks infants without evidence of severe IVH or white matter injury on CUS, 26 (86.7%) were reported to be classified GMFCS level I or II, reflecting a milder degree of CP”. Although the preterm infants with severe grades on GMFCS did have extensive grade IV IVH, this number is very small so we have been simply added the following text: The three preterm babies with GMFCS level V at 5 years all had extensive severe IVH. 

Minor issue

#4   The MRI equipment tested and its magnetic field strength should be described.

Text has been added regarding 1.5 T and 3 T (Siemens Healthcare, Erlangen Germany) see section material & methods page 2.    

#5   There is a discrepancy in the numbers. Page 3, line 130: 17 infants were imaged after NICU discharge, but in the table 2 they are 16 infants (1 + 6 + 9).

This has been corrected including text following that refers to 13 (92.8%) of the cohort. Thank you.

Reviewer 2 Report

The study aims to describe neonatal imaging frequency and results in a NICU-based cohort of children with cerebral palsy registered in the NZ CP Register.

This paper is clearly written. It conveys plain descriptive data on neonatal imaging in a selected sample with limited size. The results are relevant for clinical practice. The focus of the discussion is on the early diagnosis of cerebral palsy which goes well beyond the study aims. Further studies taking advantage of the population-base provided by NZCPR would provide a strong framework for developing clinical guidelines.

The purpose of the following comments is to improve the manuscript.

Introduction:
The concept of detection is ambiguous in the context of CP. CP is a clinical diagnosis. Neuroimaging findings contribute to interpreting the clinical presentation and history but are not part of the definition. Some predictive studies show promising results in terms of accuracy for identifying infants with high-risk for CP, not for the diagnosis of CP. Disambiguation needed.

Methods:
The authors claim to have selected a representative cohort although it is not clear what the cohort is representative of, nor do the authors provide evidence of representativeness. A flow-diagram describing the selection process –including information by gestational age groups in relation to NZCPR and ADBH NICU cohort would enhance the generalizability of the study findings.

Results:
Information on Postnatal CP cases should be provided separately - as they don’t fit in the same prediction / detection / early diagnosis context.
The main outcome measures should include confidence intervals for key findings (e.g. proportion not considered high risk, etc.). This is particularly relevant as estimates precision is limited due to sample size.

Discussion:
The interpretation of results in reference to current guidelines for the different population strata is interesting. A more thorough discussion on the availability (frequency) of neonatal imaging in the CP literature would provide a more complete picture in relation to the study objectives.
The manuscript would benefit from outlining a quality improvement exercise at the national level based on the study findings.

Author Response

Review 2

The study aims to describe neonatal imaging frequency and results in a NICU-based cohort of children with cerebral palsy registered in the NZ CP Register.

This paper is clearly written. It conveys plain descriptive data on neonatal imaging in a selected sample with limited size. The results are relevant for clinical practice. The focus of the discussion is on the early diagnosis of cerebral palsy which goes well beyond the study aims. Further studies taking advantage of the population-base provided by NZCPR would provide a strong framework for developing clinical guidelines.

Thank you for these comments. You are correct that NZCPR data is used to understand the impact of CP in NZ as well as informing development of guidelines and planning of care.  

The purpose of the following comments is to improve the manuscript.

Introduction:
The concept of detection is ambiguous in the context of CP. CP is a clinical diagnosis. Neuroimaging findings contribute to interpreting the clinical presentation and history but are not part of the definition. Some predictive studies show promising results in terms of accuracy for identifying infants with high-risk for CP, not for the diagnosis of CP. Disambiguation needed.

We understand the point that is being made in differentiating between the clinical diagnosis and the use of imaging to identify infants at risk. The wording that we use in the introduction is “ Building on the evidence that imaging and clinical assessment tools enable early detection of CP with a high level of accuracy “ and  “ using proven imaging and clinical assessment tools [11-13] to optimise early CP detection” . In both cases we are reflecting the recent shift towards incorporating imaging and other tools into a process that ends with a CP diagnosis. In fact the word detection is used in the title of references 9, 14 and 20 and highlights use of such tools that may bring the child forward earlier for clinical diagnosis. This is a relatively recent trend as these references are from 2020 onward. 

The word diagnosis could be substituted for detection in opening paragraphs if the editor preferred that option. However, the word detection was chosen to reflect the wider process leading up to that point.

Methods:
The authors claim to have selected a representative cohort although it is not clear what the cohort is representative of, nor do the authors provide evidence of representativeness. A flow-diagram describing the selection process –including information by gestational age groups in relation to NZCPR and ADBH NICU cohort would enhance the generalizability of the study findings.

The intention is to have a cohort that represents NICU Graduates who subsequently develop CP. We understand that this covers a range of gestations and pathways and that the cohort comes from a single centre. However, the two key elements relating to our aims are admission to NICU (which in addition to indicating risk, provides an opportunity to identify risk and plan monitoring or investigation) and the fact that the child subsequently developed CP. We believe our cohort to be representative of this target population given the high rate of ascertainment of individuals with CP on the NZCPR in the Auckland Region and the relative proportion of infants by GA in the NZCPR and Auckland admission data.

We have updated the text:

In the NZCPR dataset 44% of children have been identified to have a history of admission to a NICU or SCBU. Thus, to obtain a representative cohort of children, with a diagnosis of CP associated a neonatal admission, we identified ADHB NICU graduates from the NZCPR born 2000 - 2019 inclusive. ADHB NICU admissions include a cross section of gestational ages with approximately 200 babies per year born before 32weeks, 250 born at 32 to 36 weeks and more than 400 term infants.

We consider that including the data on NZCPR and NICU/SCBU admission is clear for the reader and more concise than a flow diagram

Results:
Information on Postnatal CP cases should be provided separately - as they don’t fit in the same prediction / detection / early diagnosis context.

It is agreed that postnatal events may represent an alternative pathway to CP. However, these are a complex group of babies with prolonged neonatal admissions and adverse episodes at several stages. Apropos the prediction or diagnosis being separate to a neonatal admission, this is not true for events associated with a complex neonatal course involving perioperative instability, cardiovascular collapse associated with infection or acute events related to problems with respiratory support.  The small number of infants with these adverse postnatal events have been included because of imaging within the scope of this study.

The main outcome measures should include confidence intervals for key findings (e.g. proportion not considered high risk, etc.). This is particularly relevant as estimates precision is limited due to sample size.

This is an interesting point for discussion.

In regard to the broad question of reporting estimated risk and applying this to a larger population, we agree that confidence limits are dependent on sample size. Further, we recognise that a single-centre dataset will be of a limited size. However, it is also true that some statisticians consider that when researchers have studied the whole population (i.e. the single-centre dataset) then CIs should not be used as the results are pertaining to that “whole” population. So rather than try to create confidence limits from the reported cohort, we prefer to be clear about imaging limitations in the discussion section and ensure the reader is well informed regarding imaging yield, including meta-analysis, and study in different age groups.     

To address the reviewer’s specific point, of “key findings” (e.g. proportion not considered high risk, etc.) as presented in the results section, we have amended the text to just specify the ultrasound grade of IVH rather than include a judgment about any implied level of risk. This is also consistent with the results being purely results rather than including a discussion element. The new wording is: However, a high proportion 30/53 (57%) had either no IVH or mild (grade I-II) IVH. 

Later, in the discussion section, we address detection abnormality further by stating: “Although there was a high compliance with the program, normal or only mildly abnormal ultrasound results were common in this group so would not have triggered focused follow up for CP.” Then explain further  “This is consistent with the lower sensitivity of ultrasound compared with MRI [12] and means that supplemental MRI may be required in some infants.” Further on “ Neuroimaging ………has recently been reviewed [26,27] with varying rates of abnormality reported.” Finally in discussing study weakness areas we say that “imaging technique may have an element of operator dependency …”. Overall, we consider that if the reader is aware of this range of literature it will give a good understanding of potential caveats and issues.  

Discussion:
The interpretation of results in reference to current guidelines for the different population strata is interesting. A more thorough discussion on the availability (frequency) of neonatal imaging in the CP literature would provide a more complete picture in relation to the study objectives.

We have added text regarding preterm and term MRI and generalizability.

“MRI is available in other tertiary neonatal centres in New Zealand so our findings are likely generalizable. However, it is not an unlimited resource and takes time and nursing time to perform so guidelines would assist in directing use…..”. 

The manuscript would benefit from outlining a quality improvement exercise at the national level based on the study findings.

This work will develop and promote best practice recommendations for paediatric health care professionals, which support timely diagnosis and early intervention for CP, with an initial focus on 0-2 years. The guideline document will include a section on identification of those at increased risk and promote use of the tools that assist in early CP diagnosis.  As part of implementation there will be ongoing work including education through the Cerebral Palsy network. 

Round 2

Reviewer 1 Report

The manuscript has been revised appropriately.

Reviewer 2 Report

Agree